# Unpaid caregivers' process of collaborating with others during older adult hospital-to-home transitions: A grounded theory study

**Daniel J. Liebzeit**[ID][1]*, **Saida Jaboob**[1], **Samantha Bjornson**[1], **Olivia Geiger**[1], **Harleah Buck**[1], **Sato Ashida**[2], **Nicole E. Werner**[3]

**1** The University of Iowa College of Nursing, Iowa City, IA, United States of America, **2** Department of Community and Behavioral Health, The University of Iowa College of Public Health, Iowa City, IA, United States of America, **3** Indiana University School of Public Health- Bloomington, Bloomington, IN, United States of America

* daniel-liebzeit@uiowa.edu

**Data Availability Statement:** Data may be available upon request from the corresponding author. Public deposition would breach compliance with

## Abstract

Unpaid/family caregivers provide support critical to older adult hospital-to-home transitions, but lack time and preparation. There is limited evidence regarding important collaboration for caregivers during the transition. The objective was to examine caregivers' process of collaborating with others, including other family members, healthcare professionals, and community, social, and professional networks, during older adult hospital-to-home transitions. This study utilized grounded theory methodology. One-on-one interviews were conducted with unpaid/family caregivers of an older adult during a hospital-to-home transition. Data were analyzed using open, axial, and selective coding. Participants (N = 16) relationship to the older adult included: partners (n = 8), friends (n = 4), children (n = 3), and siblings (n = 1). Most were female (n = 14) and living with the older adult (n = 10). A conceptual model was developed which illustrates participants' process through 3 stages: 1) identifying and learning the caregiver role, 2) collaborating with others to provide care and support to the older adult, while supporting themselves, during the hospital-to-home transition, and 3) supporting the older adult's progress in recovering independence or planning to provide long-term care and support. Participants described multiple approaches to collaborating with others: caring on own, caring in network, caring with healthcare professionals, and caring with social and professional networks. Implications include the need to recognize and promote utilization of care networks, as it may help address negative outcomes associated with caregiving. We also identified opportunities to further leverage caregivers' social/professional networks and increase focus on caregiver needs in healthcare encounters.

## Introduction

Older adults (age 60 and older) experience complex transitions from an acute inpatient hospitalization back to community living (i.e., hospital-to-home) [1]. Support (including medical/nursing tasks, functional, emotional) from unpaid/family caregivers is important to address

the protocol approved by our research ethics board. Please send requests by email to the corresponding author (daniel-liebzeit@uiowa.edu) and the University of Iowa Office of Nursing Research and Scholarship (nursing-research@uiowa.edu).

**Funding:** This work was supported by the Barbara and Richard Csomay Gerontology Research Award at the University of Iowa College of Nursing. The funders had no role in study design, data collection and analysis, decision to publish, or preparation of the manuscript.

**Competing interests:** The authors have declared that no competing interests exist.

their multiple healthcare, physical, and social needs [2–4]. However, unpaid/family caregivers include a wide range of individuals (family members, friends, neighbors, etc.), roles, and functions (self-care, self-management, social and emotional support) [5–8]. Further, caregivers experience challenges during hospital-to-home transitions including insufficient training and information and significant time commitment [9, 10]. Research has documented relationship strain between the caregiver and older adult [11] and increased caregiver burden [12] during hospital-to-home transitions. Caregiving during transitions often negatively impacts caregivers' physical and emotional health and functioning [13, 14]. Therefore, it is critical to understand the caregiver perspective, various approaches to caregiving, and their needs during older adult hospital-to home-transitions to improve available supports/resources and address negative outcomes.

While there are a range of qualitative studies on unpaid/family caregiver experience of older adult hospital-to-home transitions [3], there are some critical gaps. Prior qualitative research has focused primarily on caregiver involvement in healthcare interactions [15–21], inefficient discharge processes [15, 20, 22–24], and tasks (e.g., medical and nursing) related to the older adult care [17, 18, 21, 22, 24–27]. However, emerging evidence suggests a more complex collaborative process during hospital-to-home transitions [15, 28]. This more complex collaboration includes an individual caregiver and older adult, and often a larger family unit, social (friends, family, neighbors) and professional network, and various community-based resources and services. We currently know little about how caregivers collaborate with those other than the older adult and healthcare professionals during older adult hospital-to-home transitions. Underappreciation of a broader social context of individuals and families is an impediment to providing "family-centered" care that addresses concerns and needs of both the older adult and family units [29].

This study aims to address this gap by examining unpaid/family caregivers' process of collaborating with others, including other family members, healthcare professionals, and community, social, and professional networks, during older adult hospital-to-home transitions. Information gleaned from this study can inform development of hospital-to-home transition interventions that better leverage community resources, social networks, and professional services to improve both older adult and caregiver outcomes.

## Materials and methods

### Study design

This qualitative study utilized grounded theory (GT) methodology. GT is based in Symbolic Interactionism, which postulates that individuals respond to events and situations based on their understanding of those events/situations [30, 31]. GT includes a structured approach to construct an interpretive understanding of participants' experience with social process [32, 33]; this study focuses on the social process of how caregivers collaborate with others during older adult hospital-to-home transitions. It is critical to understand caregivers' broader social context during the transition to address gaps presented in the literature and advance the science toward development of effective interventions.

### Study participants

This study was approved by the University of Iowa Institutional Review Board. Participants were recruited as part of a larger GT study focused on older adults' collaboration with a support team during hospital-to-home transitions [28]. Older adult (age 60 and older) participants recruited from a large Midwestern teaching hospital (N = 25) were asked to identify a caregiver interested in completing an in-depth one-on-one interview. Caregivers were eligible if they

were: a) 18 years or older, b) identified as someone who provides support to the older adult (e.g., self-management, self-care), c) unpaid, and d) able to access a phone. Individuals were excluded if unable to provide consent or otherwise unable to share experiences following a hospitalization. Nineteen of 25 older adult participants from the parent study identified an eligible caregiver and provided contact information. Of 19 potential caregiver participants, 3 could not be reached by phone, resulting in 16 caregiver interviews. Main results from older adult interviews are reported elsewhere [28]. In this paper, we report the main results of the caregiver interviews.

## Sampling

We used convenience sampling and an iterative process wherein data collection, analysis, and sampling occurred simultaneously. The present analysis occurred simultaneously with recruitment for the larger study, and we recruited until we reached data saturation for both the larger study (older adult interview data) and the present analysis (caregiver interview data) [33, 34]. Consistent with GT we used theoretical sampling to inform sampling and modify interview questions to address gaps in the data [35, 36]. For example, as we noticed potential different approaches to collaborating with others during the transition, we targeted participants who could provide additional insight into certain approaches and conditions impacting their approach.

## Data collection

Sixteen caregivers completed a one-on-one interview conducted by DL between April 2021 and August 2022. Participants provided informed consent verbally over the phone, which was documented in their participant record. Interviews lasted 30–40 minutes and occurred by phone (due to COVID-19 contact precautions). Interviews took place approximately 30–60 days following the older adult's hospital discharge and were audio-recorded and transcribed verbatim. The interviewer took field notes, including emotions, pauses, or emphasis placed on words or phrases.

The first 5–10 interviews were relatively unstructured to allow for a rich description of participant experiences [32]. Interview guides were refined as we gained a more detailed understanding of the participants' experience and main stages and approaches of their collaborative process during the transition. The interviews began with an open-ended question to get participants discussing their transition experience, e.g., "Please tell me how things have been going since [older adult] returned home from the hospital?" Typically, participants provided detailed information about their perspective of the older adult's experience. Follow-up questions then focused on how they supported the transition and important support and resources for themselves and the older adult. Participants spontaneously identified multiple approaches to collaborating with others during the transition. Follow-up questions aimed to understand how, when, and why they used these approaches, and the various stages of their transition process. For example, when participants described how they worked with another person who also supported the older adult during the transition, a follow-up could be "tell me more about how you worked or collaborated with them".

## Data analysis

Four research team members coded each interview transcript individually, and then met weekly to discuss. We used team-based consensus discussion to resolve coding disagreements. Data were analyzed using open, axial, and selective coding [36]. Open coding included line-by-line analysis to create initial codes and group similar codes into categories [35]. We then

**Table 1. A coding example from data analysis.**

| First order findings (actual text) | Second order interpretation (open coding) | Higher order abstraction (axial and selective coding) |
|---|---|---|
| I was very on top of communicating with everybody, her doctors, and I wanted us all to be on the same page. She's got a great support system because between my brother and me and my sister-in-law, we all are on a group conversation so everybody knows so whoever's taking her next has what the other person learned. | Communicating and supporting using a group approach | Caring in network: providing care and support |
| [My sister-in-law] did take my mom to all her radiation appointments because that was every day. But if it's just when she does her three months and we try to take turns because it's so draining to be at a hospital for eight hours. | Taking turns to reduce drain | Caring in network: supporting self |

used axial coding to identify common categories across participants, dimensions of categories, interactions between categories, and conditions that impacted interactions [35]. Selective coding included identifying the core category in the data and constructing the conceptual model [35, 36]. Throughout the coding process, we used constant comparative analysis to compare participants' experiences individually and as a group, develop categories shared across participants, and compare and relate categories to develop the conceptual model [36]. A coding example from our data analysis is provided in Table 1.

## Rigor

The research team had varied clinical and research backgrounds to reduce individual bias and improve credibility of findings [32, 37] including grounded theory expertise (DL), two doctoral students with strong clinical backgrounds in caring for older, chronically ill patients (SJ, SB), and an undergraduate nursing student (OG). We provided supportive quotes from multiple participants and kept memos as an audit trail of procedures to improve credibility and reproducibility [32, 35]. Member checking during interviews helped confirm the conceptual model accurately described participants' experiences [32, 37].

## Results

Participants (N = 16) relationship to the older adult included: partners (n = 8), friends (n = 4), children (n = 3), and siblings (n = 1). Most were female (n = 14), living with the older adult (n = 10), and all were white, non-Hispanic (Table 2). A conceptual model (Fig 1) was developed that illustrates caregivers' process of collaborating with others during older adult hospital-to-home transitions through 3 stages: 1) identifying and learning the caregiver role, 2) collaborating with others to provide care and support to the older adult, while supporting themselves, during the hospital-to-home transition, and 3) supporting the older adult's progress in recovering independence or planning to provide long-term care and support. Participants described multiple approaches to collaborating with others during the transition: caring on own, caring in network, caring with healthcare professionals, and caring with social and professional networks. Conditions that impacted participants' approach to caregiving included finances, caregiver health, and older adult chronic conditions or repeated hospitalizations. Consequences varied based on participants' approach to caregiving. What follows is a description of categories across caregivers' process of collaborating with others during older adult hospital-to-home transitions (shown in Fig 1). Table 3 provides a summary of main categories with quotes.

### Stage 1: Identifying and learning caregiver role

Participants described initially identifying and learning their caregiving role as the older adult began their hospital-to-home transition. This initial stage seemed to inform the participants

**Table 2. Participant characteristics (N = 16).**

|  | Mean | Range |
|---|---|---|
| **Age (years)** | **66.38** | **38–81** |
|  | **Frequency** | **%** |
| Female | 14 | 88 |
| White | 16 | 100 |
| Relationship to the older adult |  |  |
| Partner | 8 | 50 |
| Child | 3 | 19 |
| Friend | 4 | 25 |
| Sibling | 1 | 6 |
| Living situation |  |  |
| With the older adult | 10 | 62 |
| Alone | 2 | 13 |
| With other family | 4 | 25 |
| Marital Status |  |  |
| Single | 1 | 6 |
| Married | 12 | 75 |
| Divorced | 1 | 6 |
| Widowed | 2 | 13 |

approach in later stages of their transition process, such as through information gained and relationships formed. Portions of stage 1 occurred before discharge, and some afterwards. They described multiple ways of identifying and learning their role, including involvement

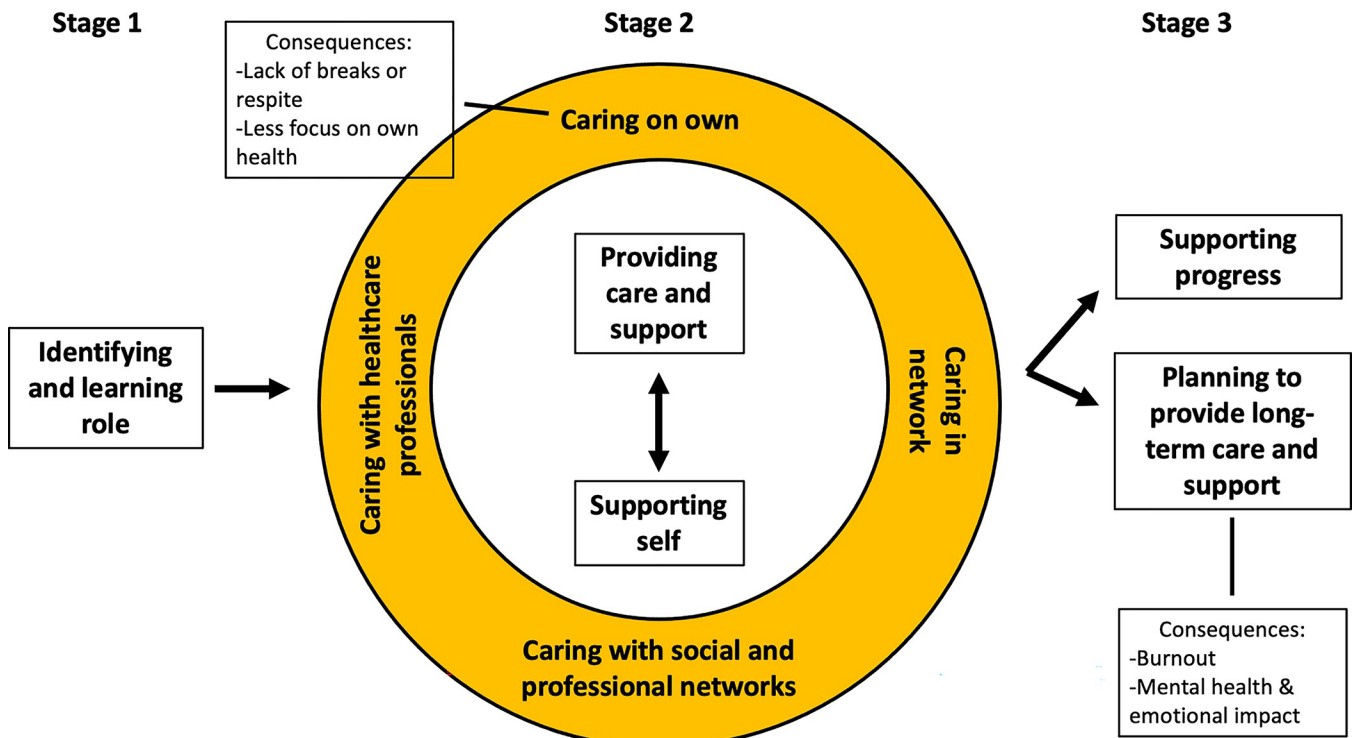

**Fig 1. Conceptual model illustrating participants' process of collaborating with others during older adult hospital-to-home transitions.**

**Table 3. Summary of main categories with quotes.**

| Category | Quote |
|---|---|
| **Identifying and learning caregiver role** | What the wound should look like. And if it was like, had an odor to it or anything like that, things that we could look for personally like oh, this is a red flag, we need to actually have her seen by a doctor, or if we had questions with the nurse or something. But otherwise, the doctors at [nearby hospital] were really helpful on things that [older adult] could do, she couldn't do, and stuff like that. And paperwork. They sent home a lot of documentation that we could read too, if we couldn't remember everything, that we could fall back on. (CG8) |
| **Caring on own**: participants took on responsibilities themselves largely without collaborating with others | We are both, pretty much, self-sufficient. We, um, just talk and, pretty much, we deal with it. (CG1) |
| **Caring in network**: multiple persons, often multiple members of the family or friends, worked together as caregivers of the older adult | [Older adult's daughter] would come in for the weekend if it was necessary. And they have a child just started as a freshman at [nearby university]. And so, they're going to have parents' weekend, the 24th, and they're coming in for that weekend, and they'll be coming and going. And usually [their son] comes in from [further state] for homecoming at the football situation. And so, this fall, there'll be several trips back and forth by her kids. And if anything, if we needed anybody, [their daughter], especially in [nearby larger city], if we needed her for some reason, she could be here in four hours. (CG5) |
| **Caring with healthcare professionals**: interactions either regarding the older adult's care or ways to support themselves as a caregiver | It's definitely a positive thing to have the nurse come in, even if it was just once a month or twice a month, whatever it was. I guess it put her at ease too a little bit with the nurse, talking positive to her, letting her know that everything was going good and her process was up to standard, and everything was looking okay at that time. I guess that was all real good backing for her so that was a positive step, having her come in when she'd come in. (CG3) |
| **Caring with Social and Professional Networks**: one primary caregiver getting support by leveraging social and professional networks as needed | I got good neighbors, I live in a good neighborhood and my neighbor come over a couple of times and mowed my lawn for me. He'd come over and check on us, make sure we're all right. (CG3) |
| **Supporting Progress** | Well, [older adult] was getting up sometimes when I was there and she's like, "I'm fine, I can do that," and then it was just like she was more independent, she didn't need me there. Then she was functioning very well, and you could tell that she was going back to her old self. She could get around by herself, she could do a lot of. . . She was doing the steps by herself. So it's not like she couldn't do anything on her own. (CG8) |
| **Planning to Provide Long-term Care and Support** | That'd be a tough [if older adult wasn't independent], but I would say that it's got to be more of a home environment. No going into a care center or anything. If I had to I'd hire somebody to come help. That's just the way I feel he would better be able to adapt to a situation. (CG11) |

during hospitalization, discharge education, and information from the older adult. Some felt they received inadequate information during the hospitalization and discharge process.

"I didn't know how [the wound] should look and I didn't know that [older adult] was supposed to be taking a shower, we were supposed to be washing it off. So we didn't. . . So, I wish we had more like, "These are things to look for if it's going bad and these are just

normal things. So, if you see these, these are expected and don't worry about them." I think that would have been helpful. And I didn't know what questions to ask." (CG2)

If participants were not involved during hospitalization or discharge, they were reliant on the older adult to provide information as they returned home. The way in which participants identified and learned their role seemed to inform their approach to collaborating with others to provide care and support, while also supporting self, in stage 2.

## Stage 2: Collaborating with others to provide care and support while also supporting self

Participants described types of care and support provided to the older adult during the transition. These included physical/basic needs, medical tasks, care coordination, advocating, emotional mental support, social engagement, taking over household responsibilities, identifying resources/services, and monitoring safety. Some participants provided more time-consuming daily support. Others checked in once a day to every few days, such as when they didn't live with the older adult and there was less need for monitoring throughout the day. Some participants described strategies for supporting themselves, given the demands of the caregiver role, but it was indicated as a lower priority than supporting the older adult. Participants described multiple approaches to proving care and support and supporting self: *caring on own*, *caring in network*, *caring with healthcare professionals*, and *caring with social and professional networks*. Some participants used the same approach throughout the transition, while others used different approaches based on timing and nature of their need.

**Caring on own.** Caring on own meant participants took on responsibilities themselves largely without collaborating with others.

*Providing care and support*. Many participants approached providing care and support to the older adult on their own. This was due to participants either feeling self-sufficient or a sense of responsibility to the older adult. Some were hesitant to express potential concerns to healthcare professionals and instead internalized the challenges. While many participants who approached caregiving on their own had family nearby to help, they still elected to manage on their own. One participant described trying to manage on their own despite health and financial challenges.

"[Older adult] will take care of me, and I'll take care of her. I do that whether I'm sick or not [. . .] we may be broke as shit or we may not have gas money for this or that, different things like that, but we make it. I don't ask my kids for no help." (CG3)

A few participants attributed it to not wanting to be a burden on their children or other family or friends. Those with family members living out of state described increased obligation on person(s) in proximity. Some had friends or members of their church offer food or other support.

"I had good support in my church as well and people, 'what do I say? What can we do? Can we bring you food?' [. . .] And I don't really know if there was anything that anybody could do, we didn't need food and we couldn't have company. So I think it's sadly, it's kind of something you have to just deal with on your own." (CG7)

Participants' perception of support varied in terms of whether it met their needs and their level of openness to accepting help.

*Supporting self*. When caregiving on their own, participants often described learning to deal with the extra stress and demand of caregiving.

"I guess I was so used to it that there was maybe a certain amount of stress, basically, just connected to worried about him. I kind of learned to deal with it." (CG11)

In some cases, participants described getting away to engage in activities they enjoy or socialize with others as a way of supporting their own well-being.

"I just sometimes slip out and take a walk or go down to the gym and walk. Just to get myself back into being away a little bit. I try not to miss anything with my grandkids. I never have. But I go alone. Anyway, it's just hard for him to go out. [...] I've been a widow for a long time when it comes to going to events. I just go by myself." (CG13)

In other cases, they and the older adult may confide in each other as a way of supporting themselves. This approach could help strengthen their relationship in ways the caregiver found meaningful.

**Caring in network.** Caring in network meant multiple persons, often multiple members of the family or friends, worked together as caregivers of the older adult.

*Providing care and support*. When collaborating in a care network, participants described ways they coordinated their efforts, such as by dividing tasks inside and outside the home, or by time of day. This included dividing responsibility to visit during healthcare appointments or hospitalizations, wherein one caregiver could keep other individuals informed.

"Well, [older adult's] sister lives down the street from us too. And her niece took her to appointments [...] next week she's got them back-to-back, two days in a row. So I'm going to take her one day and somebody from her work is going to take her the next day. So, we kind of spread it. We're very, very lucky." (CG2)

While some actively coordinated and divided tasks, in other cases one person may act as the primary person while another served as a backup. Quality of communication amongst care networks varied. Some participants described clear strategies to communicate effectively and stay on the same page.

"I was very on top of communicating with everybody, her doctors, and I wanted us all to be on the same page. She's got a great support system because between my brother and me and my sister-in-law, we all are on a group conversation so everybody knows so whoever's taking her next has what the other person learned." (CG8)

Some noted the importance of meeting as a group, in addition to staying in contact by phone or text. Other participants described limited contact between themselves and other members of the care network, especially when one person was a family member and another a friend. This seemed to decrease potential coordination of the care network. Care networks were noted as especially important when one caregiver may be prone to declines in health or function. Having a person with healthcare experience was noted as another potential benefit.

*Supporting self*. Participants "caring in network" described other caregivers as allowing them to take breaks and have their own lives. Having other caregivers available helped the older adult to also encourage one caregiver to take a break. This was particularly important

when the older adult had many appointments or other intense demands that would be difficult for a single person to manage.

> "[My sister-in-law] did take my mom to all her radiation appointments because that was every day. But if it's just when she does her three months and we try to take turns because it's so draining to be at a hospital for eight hours." (CG8)

Care networks also appeared to result in more of a shared venture, which increased camaraderie and feelings of being supported by others.

> "I just think that we're a very fortunate group that we have shared interests and have known each other and worked with each other for such a long time. That it's just a great relationship and support. Everyone should have one. Everyone should have a posse." (CG12)

It also appeared to facilitate them asking others for help, as they were acquainted to working with and reaching out to others, versus approaching caregiving on their own.

**Caring with healthcare professionals.** Caring with healthcare professionals meant participants interacted with them either regarding the older adult's care or ways to support themselves as a caregiver.

*Providing care and support.* Many caregivers felt highly reliant on healthcare professionals (e.g., physicians, nurses) to provide them information necessary to support the older adult. However, participants who were less involved during the index hospitalization or post-discharge follow-up visits felt a missed opportunity to gather information, ask questions, and develop that relationship.

> "I, unfortunately, wasn't really involved in [conversations with healthcare professionals] much. It was more my father and my mother. I wish I had been more involved, because I feel like I could answer questions." (CG4)

Caregivers who were employed described challenges in attending follow-up appointments due to work schedules and ability to take leave. Participants described collaboration with healthcare professionals as particularly challenging when they didn't have an existing relationship. Some participants described frustrations with limited information provided about what could go wrong, as well as healthcare professionals not listening to their input and concerns. Even when concerning symptoms did emerge, some participants were unsure when and where to reach out or what questions to ask.

> "I just didn't know who to call for mostly for his physical things. Like one at one point the wound in the front, I think it was connecting to the pocket of an infection. [. . .] I mean, we would change that bandage and it would just be full of crap, looked like snot or something. And there were times when I didn't know what to do with that. [. . .] I guess you don't know what you don't know." (CG7)

In some cases, challenges knowing who to ask arose out of having many different healthcare professionals involved in their care and no clear direction of who to contact. However, when healthcare professionals did listen to participants input and appreciate their advocacy, it could result in improvement in care.

"And because my mom was so loopy [. . .] I tried to have my mom tell [her physician] what she was doing for medication for her leg that hurt so bad, and she couldn't even tell. So, I stepped in. I'm like, "Hey, [physician]. This is [participant's name], I just want to let you know what I'm seeing," and he could even tell she was disoriented because she's taking so much medicine. [. . .] We had great communication." (CG8)

Whether participants felt involved in conversations about the older adult's care varied. A few participants felt that their perspective or questions were actively sought and valued. Others felt they needed to push or advocate for their involvement in conversations. When participants had caregiver access to the patient portal, it could also help to facilitate their collaboration with healthcare professionals. Whereas lack of access to patient records was a barrier for others.

Participants described additional healthcare services such as inpatient rehabilitation and home nursing care as important extra layers of support from healthcare professionals during the transition. However, a few participants felt they had unfair responsibility to advocate for rehabilitation services, where their healthcare professionals should have identified a need for referral. Participants who had support from home nursing expressed many benefits, including increased comfort in having someone to ask questions, check on progress, or identify any concerns.

"We were offered a visiting nurse or someone to come in, but I figured I could do it. [. . .] And there was maybe one point where I wished someone would've been able to just pop in and take a look at it to see if what we were doing is right." (CG7)

A few who passed on home nursing care later recognized ways it could have been helpful.
*Supporting self*. Participants' collaboration with healthcare professionals was focused on providing care and support for the older adult. Their input could give assurance that they are doing the right things.

"I think just talking to the staff there and the nurses. It was comforting to talk to them because they knew the right things to say and that helped. Well, I think they just kind of reassured me." (CG10)

Upon asking, participants did not identify ways that any healthcare professionals gave input, advice, or means of supporting themselves as caregivers.

**Caring with social and professional networks.** Caring with social and professional networks meant participants engaged their social networks, professional services, and other community resources to provide care and support while supporting themselves as caregivers. This was distinct from caring in network in that there is only one primary caregiver getting support by leveraging social and professional networks as needed.

*Providing care and support*. Family and friends were often the first layer of additional support considered for the older adult. Participants were often hesitant to have other strangers in the home, despite the help they could provide. Further, participants were often hesitant to pursue resources potentially available in their community, due to feelings that they or their care network could handle things on their own. Others were unaware what services may be available to them in their area without paying out of pocket.

"The question I've always had, what's available to people in the home? It just seems like the government wants you to leave your home. Let someone else take it over. Pay the taxes and

shove you in a nursing home or assisted living. What is available out there that you can have come in your home without it costing me?" (CG13)

Those with a professional background in healthcare could utilize their professional networks for medical, nursing, or other advice.

*Supporting self*. Participants often described friends and neighbors as important for helping with the additional responsibilities that caregivers may need to take over from the older adult during the transition. When participants had to travel to support an older adult, friends and neighbors were also important to check on and maintain their home while they were away. However, some participants preferred paid professional services to relieve some of the extra responsibilities placed on them during the older adult's transition.

"We do have a guy that comes and scoops the snow. Snow blows it. If I can't get the back lot mowed, we do have the same guy that'll come and mow it. That's all paid stuff though. I don't let anybody work for us unless I pay them." (CG13)

These participants preferred not to overly rely on family and friends for unpaid help.

While some participants leaned on family and friends as a support group to talk to and seek support and guidance, others hoped to identify more formal support groups with shared experiences.

"But maybe if there was, like, some support groups or somewhere where you could go and you could visit with [...] I know everybody's different but, you know, it's kind of like you go into it blind." (CG1)

Participants who were employed outside the home may use their work relationships as an emotional outlet.

## Stage 3: Supporting progress versus planning to provide long-term care and support

**Supporting progress.** Through their hospital-to-home collaborative process, participants described a desire to ultimately support the older adult's progress towards improved health and independence. In the later part of their transition experience, participants described their role in stepping back to encourage their independence in activities.

"As soon as she's able to do these things, then we let her do those things again because it's important that she keeps up her routine and is active as much as she can be." (CG5)

Participants moved to supporting progress either after a certain amount of time passed or relied on cues from the older adult for when it was time to step back.

Participants who had support from physical therapy or other healthcare professionals also sought and used their input to determine when and how to encourage the older adult's independence. Some participants had to encourage the older adult to have patience through difficult parts of their recovery.

"I'm the wife. Now, if he's at the hospital or therapy, he listens to [them]. He listens to those people. But you don't listen to your mate." (CG13)

Input from healthcare professionals was particularly important for participants whose older adult may be reluctant to follow their own encouragement.

**Table 4. Conditions impacting participants' approach to caregiving during transition.**

| Condition | Definition | Quote |
|---|---|---|
| Finances | Finances impacted access to professional services, such as support in the home that could provide respite for participants. | Everything costs. If you want somebody to come to the house, that cost. I don't even know how to get that done. (CG13) |
| Health of Caregivers | Those worried about their own declines in health or function were more likely to consider caring in network or caring with social and professional networks. If using a care network, health may also influence who becomes primary versus secondary on that team. | [Her husband has] got plenty of his own health things going on right now, so it's hard for him to try to manage himself and her, which is why I've kind of, I guess, stepped into the role of taking care of her. (CG16) |
| Older adult with a chronic illness or repeated hospitalizations | This increased difficulty on participants helping the older adult regain independence. Therefore, they may spend more time in the hospital-to-home collaborative process, which could contribute to burnout of themselves, their networks, and increased costs. Alternatively, it may lead to them more likely planning to provide long-term care and support than supporting the older adult's progress toward independence. | [Older adult has] had COPD for a long time. I'd say this has been going on for 20 or more years. Congestive heart failure. Now he was doing real good. He was on oxygen a decade or more ago, two decades ago maybe. Got rid of all of it. Was just doing great. Then all of a sudden, it happens again. Now all the oxygen stuff is back. I don't know, it's just a part of life. (CG13) |

**Planning to provide long-term care and support.** Some older adults did not make significant progress in recovering their independence during the hospital-to-home transition process. Instead, their caregiver described planning to provide long-term care and support. They described continuing to collaborate with others to manage ongoing dependence of the older adult.

"He's very dependent on me and my mom. He's pretty much house bound. He doesn't get out. He's on oxygen and the oxygen tanks are too heavy for him to carry." (CG4)

Participants often focused on trying to keep the participant from further changes, such as keeping them at home as long as possible. Some participants' collaboration with others shifted as they considered relocating the older adult to live with another family member, assisted living, or other care facility.

"It was like would she be better off in a facility or assisted living? [. . .] I even stopped by one to talk to somebody, kind of get pricing and an idea of what's all involved in their facility. But I do think that until they're ready, my mom and dad, something like that's not going to happen." (CG16)

However, another challenge was having the older adult agree to changes in their living situation.

**Conditions.** Conditions provide the context that impacted participants' approach to caregiving and collaborating during the transition. Conditions that impacted participants' approach to caregiving included finances, caregiver health, and older adult chronic conditions or repeated hospitalizations (Table 4).

## Consequences

Participants described consequences of their transition process differently based on their approach to caregiving and whether they were able to support progress versus planning to provide long-term care and support (see Fig 1).

**Caring on own.** Participants who were caregiving on their own described multiple consequences or outcomes from the lack of collaboration and resulting poor self-care, including having no breaks and often giving up activities that were important to them.

"I used to go to church. But I don't know. I'm so wore out. I don't get up in time to get there. I guess I should go down to [nearby town] on Saturday night. But I haven't been to church since Easter. [. . .] I've got a lot of good friends at church. (CG13)

Lack of breaks or respite when caring on own contributed to participants' burnout and frustration.

"I'm trying to be the wife, the caretaker, hard knock taker. If there's anything I've got in this life, it's a lesson in hard knocks. I've got my PhD in hard knocks." (CG13)

They also described less time and ability to focus on their own health.

"I do have some medical problems that the doctors would like to be a little more aggressive with and I won't let them, but as a rule, I don't like [older adult] to be stressed. And so, I try to not need as much help." (CG11)

This could contribute to delays in treatment or care seeking that may compromise the participant's health and ability to support the older adult.

**Planning to provide long-term care and support.** Participants planning to provide long-term care and support described consequences of long-term caregiving.

"If you're a long-term caregiver, I can see where it's wearing. [. . .] I didn't even think about it before this happened. Because I didn't have any clue that it was going to last this long." (CG2)

Often these individuals didn't feel prepared for providing support long-term and were worn out.

"I think the biggest thing is [older adult is] pretty much housebound now. And I think that's the hardest part." (CG4)

They also described challenges coping with declines observed in the older adult.

## Discussion

Our study examined unpaid/family caregivers' process of collaborating with others, including other family members, healthcare professionals, and community, social, and professional networks, during older adult hospital-to-home transitions. Our findings reveal a multi-stage transition process as caregivers support older adults through hospital-to-home transitions. Participants described various approaches to providing care and support, while supporting themselves, during the transition, which appear to involve more collaboration and utilization of networks than described in prior research. While prior qualitative research on older adult hospital-to-home transitions has focused on the role of a single caregiver undergoing the transition process [15–27], our findings suggest a more complex team or "care network" structure of caregiving. This is consistent with caregiving research in other populations and efforts to reconceptualize caregiving to appreciate various roles, individuals, and dynamics in caregiving and caregiving networks [5–8]. Three important, novel findings from this study elucidate: 1) the collaborative relationship between caregivers and healthcare professionals, 2) opportunities to leverage social and professional networks, and 3) conditions that may impact caregivers' experience during the transition.

Our findings expand prior research on caregivers and healthcare professional collaboration. Prior research has indicated a need for improved communication between healthcare professionals and caregivers and support during the transition [15, 17, 19, 26]. Our findings elucidate this issue, indicating that healthcare professional interactions with caregivers may primarily

focus on how to address older adult needs during transitions. Conversations do not sufficiently address the support needs or challenges of caregivers. A recent study supports that healthcare professionals should consider the needs of caregivers when planning interventions to address caregiver burden and promote their health and wellbeing as part of person or family-centered practice [38]. In fact, the CARE Act requires hospitals in most states to: 1) record the name of the family caregiver on the medical record, 2) inform the family caregivers of discharge, and 3) provide the family caregiver with education and instruction of the medical tasks they will need to perform for the patient at home [39]. Given caregivers' tendency to prioritize the older adult's needs over their own, counseling services may aid caregivers in recognizing their need for self-care and in helping them to create strategies that highlight their abilities and available resources [40].

Our findings also reveal opportunities to leverage social and professional networks, to support transition tasks for the older adult, and address caregiver needs. Prior studies have indicated challenges in caregivers receiving assistance from family and friends, formal support services being implemented too late or ineffective, and lack of familiarity with available community services [17, 18, 21, 22, 24]. Our findings indicate that social networks and professional services can serve an integral role in helping caregivers take on the many caregiver responsibilities while maintaining other roles, e.g., professional, social, household. There is an opportunity to further leverage these and other community resources in future research and intervention, such as more purposeful matching of those in need with appropriate aging and caregiving resources [27, 41–44]. Communities have an opportunity to take action to support caregivers such as through volunteering programs providing respite care; one example is an evidence-based program to support family caregivers of dementia patients [45]. These findings also have implications for healthcare professionals, such as the need to understand the benefits that social/professional networks can provide caregivers, identifying opportunities to ask about these networks, and referring to community resources when there is an identified need.

Third, our study identified several conditions that may impact caregivers' experience during transitions. They indicate a need to focus specifically on caregivers of an older adult undergoing multiple repeated admissions and with serious chronic illness. We must also assess the person's financial situation and other social determinants of health that impact their transition [46]. Identification and discussion of potential barriers and facilitators to an effective and collaborative approach to caregiving during healthcare encounters may provide a basis for providing and tailoring support and services to the caregiver during the hospital-to-home transition.

## Limitations

The study's design relied on retrospective report of participants' experiences. Interviews took place approximately 30–60 days following the older adult's hospital discharge, which allowed for a strong respective report of the entire transition experience. Variation in exact timing of interviews is both a strength and limitation, as it increases variation in responses that can contribute to both richness of data and variation in recount of experiences between participants. We did not interview any participants in the first 2 weeks post discharge (due to potential burden and required time to follow up and schedule the interview), which could be considered a limitation. Conduct of this study during the COVID-19 pandemic likely impacted study findings, such as access to resources and services, interactions with others, and visitation during hospitalization, which should be considered.

The parent study recruited participants from one healthcare system [28]; further study is necessary to determine how findings may be transferable to older adults and their caregivers

in other regions and healthcare systems. Our sample is relatively small and lacks ethnic, cultural, and racial diversity. It will be critical to further explore how care networks and their perceived impact/utility may differ substantially such as between individualist and collectivist cultures. Some demographic variables were not collected, including employment status, income, or socioeconomic status, and should be considered in future research. We did not include caregivers of critical care patients, those with cognitive impairment that would interfere with their ability to consent, or those transitioning to other settings such as skilled nursing facilities.

## Conclusions

Implications of this research include the need to recognize and promote care networks as opposed to focusing primarily on the role of an individual caregiver. Care networks, including multiple family member or friends, appeared to help address many of the negative consequences commonly associated with caregiving for an older person, such as caregiver health, burden, or strain. Further, our participants indicated that interactions with healthcare professionals currently focus on how to address needs of the older adult and do not sufficiently address the support needs or challenges of caregivers. There is an opportunity to better incorporate caregiver needs in models of "family-centered" care as caregivers are often hesitant to focus on their own needs. We identified opportunities to further leverage caregivers' social and professional networks; networks and caregiver access to community resources are important areas of future research and intervention.

Our findings underscore the importance of assessing and considering contextual and situational factors when trying to support caregivers, and this must be further investigated in larger and more diverse samples. We must appreciate various barriers to caregiving, including caregivers' health and financial situations, as well as disparities in access to community resources, such as for those in rural areas. Recent policy efforts such as the CARE act [39] provide a foundation on which to bolster and tailor support and resources provided to caregivers of hospitalized older adults.

## Author Contributions

**Conceptualization:** Daniel J. Liebzeit.

**Data curation:** Daniel J. Liebzeit, Olivia Geiger.

**Formal analysis:** Daniel J. Liebzeit, Saida Jaboob, Samantha Bjornson, Olivia Geiger, Harleah Buck, Sato Ashida, Nicole E. Werner.

**Funding acquisition:** Daniel J. Liebzeit.

**Investigation:** Daniel J. Liebzeit, Saida Jaboob, Samantha Bjornson, Olivia Geiger.

**Methodology:** Daniel J. Liebzeit.

**Project administration:** Daniel J. Liebzeit.

**Resources:** Daniel J. Liebzeit.

**Supervision:** Daniel J. Liebzeit.

**Validation:** Daniel J. Liebzeit.

**Visualization:** Daniel J. Liebzeit, Saida Jaboob, Samantha Bjornson, Olivia Geiger, Nicole E. Werner.

**Writing – original draft:** Daniel J. Liebzeit.

**Writing – review & editing:** Daniel J. Liebzeit, Saida Jaboob, Samantha Bjornson, Olivia Geiger, Harleah Buck, Sato Ashida, Nicole E. Werner.

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
