## [Decision Letter · Decision Letter 0]

12 Jun 2024

PONE-D-23-42012Unpaid Caregivers’ Process of Collaborating with Others during Older Adult Hospital-to-Home Transitions: A Grounded Theory StudyPLOS ONE

Dear Dr. Liebzeit,

Thank you for submitting your manuscript to PLOS ONE. After careful consideration, we feel that it has merit but does not fully meet PLOS ONE’s publication criteria as it currently stands. Therefore, we invite you to submit a revised version of the manuscript that addresses the points raised during the review process. Your manuscript has been evaluated by three reviewers, and their comments are appended below. The reviewers have collectively identified a number of concerns to address and opportunities to refine your manuscript. The reviewers' concerns relate in general to the reporting of the study methodology and results, but also to contextualization of this study; both in terms of this being related to your previous work and also how this work and its findings relate to other work in this field. Please ensure you address each of the reviewers' comments when revising your manuscript.

We look forward to receiving your revised manuscript.

Kind regards,

Hugh Cowley

Staff Editor

PLOS ONE

 [This work was supported by the Barbara and Richard Csomay Gerontology Research Award at the University of Iowa College of Nursing.].  

3. In the online submission form, you indicated that [Data may be available upon request from the corresponding author.]. 

Additional Editor Comments (if provided):

Reviewers' comments:

Reviewer's Responses to Questions

**Comments to the Author**

1. Is the manuscript technically sound, and do the data support the conclusions?

Reviewer #1: Yes

Reviewer #2: Yes

Reviewer #3: Yes

2. Has the statistical analysis been performed appropriately and rigorously? 

Reviewer #1: N/A

Reviewer #2: Yes

Reviewer #3: N/A

3. Have the authors made all data underlying the findings in their manuscript fully available?

Reviewer #1: No

Reviewer #2: Yes

Reviewer #3: No

4. Is the manuscript presented in an intelligible fashion and written in standard English?

Reviewer #1: Yes

Reviewer #2: Yes

Reviewer #3: Yes

5. Review Comments to the Author

Reviewer #1: Thank you for the opportunity to review this paper.

Introduction. The introduction establishes the reason for the study. Yet, the objective included in the abstract: “The objective was to examine caregivers’ process of collaborating with others during older adult hospital-to-home transitions.” The aim in the intro, are slightly different” This study aims to address this gap by examining unpaid/family caregivers’ process of collaborating with others, including other family members, healthcare professionals, and community, social, and professional networks, during older adult hospital-to-home transitions”. Consistency would help the reader.

My biggest concern is that the central justification for the paper, is based on two self-citations. Valuable though this previous work is, the authors should note, that the current paper, builds on previous work by same (almost identical) research team. It is only in the methods that a larger study is mentioned. The text on page 2. with the self-citations in full:

Prior qualitative research has focused primarily on caregiver involvement in healthcare interactions, inefficient discharge processes, and tasks (e.g., medical and nursing) related to the older adult care (3). However, emerging evidence suggests a more complex collaborative process during hospital-to-home transitions (11).

3. Liebzeit D, Jaboob S, Bjornson S, Geiger O, Buck H, Arbaje AI Ashida, Sato ; Werner, Nicole E. A scoping review of unpaid caregivers’ experiences during older adults’ hospital-to-home transitions. Geriatr Nur (Lond). 2023;53:218–26.

11. Liebzeit D, Geiger O, Jaboob S, Bjornson S, Strayer A, Buck H, Werner, N. . Older Adults’ Process of Collaborating with a Support Team during Transitions from Hospital to Home: A Grounded Theory Study. The Gerontologist. 2023 Jul 12;gnad096.

This brings me to the lack of an informative, well-structured lit review. The literature covered in the introduction may be sufficient for that task. However, as a reader, I want to know how the current study is contextualised within current literature about collaboration between unpaid/family/friend caregiver and paid/unpaid others in relation to older adults in/leaving hospital. Some explanation/definitions of informal/unpaid caregivers would be helpful. Including research about single/primary and group/family caregivers would add depth to the paper. It may also be useful to look at existing literature around caregiver self-care during transitions of the cared-for person e.g., home to hospital; home to residential care; discharge to home.

Methods:

It would be educative for the authors to provide an example from their data analysis to demonstrate the analysis process.

A table listing the main themes and subthemes, with examples of quotation would provide a summary of the findings.

Findings/Results:

The data is rich. The themes and subthemes are well-structured. It would be best to end each theme/subtheme with an interpretative comment, rather than with a participant quotation. This may mean rearranging rather than rewriting relevant theme/subthemes.

Discussion: The discussion needs development.

Please restate the purpose and summarise the key findings and then discuss each key finding in relation to the objective and relate these findings to prior studies. The authors need to show how the findings are consistent or contradictory with prior research – and as well as their own work. Where the results are contradictory, explain the possible reason.

The limits of self-citation are evident in this section. The opening paragraph begins with, “While prior qualitative research with this population has focused on the role of a single caregiver undergoing the transition process (3), our findings suggest a more complex team or “care network” structure of caregiving.” (p. 20). As noted earlier, reference (3) is a self-citation not acknowledged as the authors’ prior work. The results should be situated within the broader literature to support the authors’ claims. For example, such literature should include previous studies that critique and unpack the “primary caregiver” model.

Overall, this paper has the potential to be strong because the data is rich. A major downfall is the failure to adequately situate the study, and the results of the study, with the extensive existing literature about informal/family/unpaid caregivers' collaboration among themselves and with formal providers.

Reviewer #2: Summary:

The authors of this study sought to use grounded theory methodology to characterize and explore the process that unpaid/family caregivers undergo to collaborate with other people in order to support the hospital-to-home transition of an older adult. Interviews from 16 caregivers of older adults informed the development of a conceptual model that maps that process that caregivers undertake in the hospital to home transition. Findings identified the key role that collaboration plays in supporting this transition, and caregivers highlighted the role of family, friends, healthcare providers and larger social/professional networks in this process.

Strengths and Weaknesses

- The authors provide a strong justification for the need to explore caring through this network lens, and the need to intervene with support of caregiver’s network in order to support their caring role.

- The choice of grounded theory methodology is well supported throughout the methods,

- Methods- Line 101 page 4: This line is confusing. The authors should clarify if data saturation was reached for the parent larger study, or if the present analysis was a subset of individuals from the larger study and if data saturation was not used to decide when to end enrollment for the present analysis.

- Methods - Data analysis and credibility is well written.

- The three stages (identifying + learning, collaborating, supporting progress/planning) are well described and provide an interesting overview of the touchpoints with healthcare and community throughout this process. Seeing as there are subheadings under each stage, a summary paragraph would be helpful before proceeding into the next stage.

- Results – It is unclear in the results section how the ‘conditions’ and ‘consequences’ are intended to be woven into the three stages. Figure 1 attempts to indicate where these intersect with the stages, but more discussion of how conditions and consequences impact each stage of the transition is warranted within the text.

- Discussion – line 456 + 464-466. This work also identifies an important implication for healthcare providers, namely in understanding the benefits that social/professional networks can provide caregivers, asking about these networks and referring to community resources when there is an identified lack. There is some discussion of promoting the role of care networks, but there is room to further expand on the role that healthcare plays in helping caregivers access these networks.

- One the major limitations of this work is the lack of cultural diversity. This is referenced in the limitations section; however, the authors should consider exploring this limitation further. Care networks and their perceived impact/utility may different substantially between individualist and collectivist cultures.

Minor Comments

- The authors could comment on the later timeline (30-60 days) of post-discharge interviews and how this might have impacted responses and themes as well as whether this was a strength or limitation of their study.

- The authors should comment on how the study taking place during the COVID-19 pandemic may have affected study findings. For example, in many parts of the world, there were restrictions imposed to caregivers coming to hospital due to public health restrictions which may have affected the experiences of caregivers.

- It would be helpful for the reader if the authors were to delineate between ‘care in network’ and ‘care in social/professional networks’ earlier on. This comes up in the results section for ‘caring in social/professional networks’, but would be helpful to understand when describing ‘care in network’. Perhaps renaming to ‘shared caring in network’?

- In Table 1, sharing the employment status of participants would provide context into their capacity to provide care. Caregiver employment is referenced in Line 279 so the authors may have this data to share.

- Results - Lines 169 – 173. This section indicates first that caregivers learned their role during the transition, and then goes on to identify different ways of learning about their role, including during hospitalization. Would learning about their role not happen during the hospitalization, before the transition? Experiences which may have occurred while in hospital would be important to clarify here.

Reviewer #3: This paper reports a qualitative study looking at the role of caregivers during older adults' hospital to home care transitions. This is an important area to explore and this paper adds to the current literature. In particular, it offers a 3-stage process of what caregivers may go through, and some of the associated needs/gaps, and influencing factors. These can be tested in the future and may be addressed in future interventions.

MAJOR COMMENTS:

1. A relatively small sample size and does not appear to be diverse. This is appropriately stated as a limitation. I do not believe there is anything else the authors can do to address the limitations with the current data set (unless they plan to pursue further data collection).

2. As this study has its recruitment linked to a parent study (and given the relatively small sample size and lacks diversity), it would be important to give the readers some sense about what the parent study is about (who may be recruited to the parent study, or whether the parent study is descriptive or interventional). This will give the reader further context to interpret the data and claims made. The authors could provide more information about this in the manuscript.

3. The authors also did not give any sense of the amount of care (can be a range) provided by caregivers in the sample. This may be helpful for the readers to understand findings (e.g. one can imagine whether caregiver can care on own vs with a network may depend on the nature of tasks and amount of care required). I suggest the author make some comments about the amount of care caregivers are providing in a sample (a range would likely be needed I suspect).

4. For stage 2, it is unclear whether the categories (caring on own, caring with network, etc...) are mutually exclusive. I would also imagine that regardless of whether one is caring on own or with network, that they will interact with healthcare professionals (so unclear to me what type of caregiver is included/excluded in the caring with healthcare professionals category). Also, I imagine that any caregiver can access social and professional network (so I am unsure whether this implies in the data, there is a group of caregivers who primarily do this - but not caring on own or with network?)?? I think this requires further clarifications in the manuscript.

MINOR COMMENTS

5. The phrases "caring with network" and "caring with social and professional networks" are not self-explanatory. The authors did explain it in the middle of the manuscript (but did make several references to these concepts prior). Would recommend explaining these concepts earlier on in the manuscript.

6. Line 114: I think the author meant first :10-15 minutes (and not interviews)??

6. PLOS authors have the option to publish the peer review history of their article (what does this mean?). If published, this will include your full peer review and any attached files.

Reviewer #1: No

Reviewer #2: No

Reviewer #3: No

---

## [Author Response · Author response to Decision Letter 0]

24 Jun 2024

Thank you for the thoughtful review and detailed opportunities to strengthen our manuscript. Each of the reviewer comments have been enumerated and addressed below.

Editorial staff comments:

1. All PLOS journals now require all data underlying the findings described in their manuscript to be freely available to other researchers, either 1. In a public repository, 2. Within the manuscript itself, or 3. Uploaded as supplementary information. This policy applies to all data except where public deposition would breach compliance with the protocol approved by your research ethics board. If your data cannot be made publicly available for ethical or legal reasons (e.g., public availability would compromise patient privacy), please explain your reasons on resubmission and your exemption request will be escalated for approval.

RESPONSE 1: Unfortunately, public deposition would breach compliance with the protocol approved by our research ethics board, which is why we indicate data may be available upon request from the corresponding author.

Reviewer: 1

2. Introduction. The introduction establishes the reason for the study. Yet, the objective included in the abstract: “The objective was to examine caregivers’ process of collaborating with others during older adult hospital-to-home transitions.” The aim in the intro, are slightly different” This study aims to address this gap by examining unpaid/family caregivers’ process of collaborating with others, including other family members, healthcare professionals, and community, social, and professional networks, during older adult hospital-to-home transitions”. Consistency would help the reader.

RESPONSE 2: Thank you for catching this inconsistency. The abstract has been revised to match the statement in the introduction.

3. My biggest concern is that the central justification for the paper, is based on two self-citations. Valuable though this previous work is, the authors should note, that the current paper, builds on previous work by same (almost identical) research team. It is only in the methods that a larger study is mentioned. The text on page 2. with the self-citations in full: Prior qualitative research has focused primarily on caregiver involvement in healthcare interactions, inefficient discharge processes, and tasks (e.g., medical and nursing) related to the older adult care (3). However, emerging evidence suggests a more complex collaborative process during hospital-to-home transitions (11).

RESPONSE 3: Thank you for pointing this issue out. In the original submission, we attempted to summarize prior literature by referencing our in-depth scoping review of the literature. We have revised to include citations of primary studies and provide a more comprehensive review of current evidence: “Prior qualitative research has focused primarily on caregiver involvement in healthcare interactions (11–17), inefficient discharge processes (11,16,18–20), and tasks (e.g., medical and nursing) related to the older adult care (13,14,17,18,20–23). However, emerging evidence suggests a more complex collaborative process during hospital-to-home transitions (11,24).”

4. This brings me to the lack of an informative, well-structured lit review. The literature covered in the introduction may be sufficient for that task. However, as a reader, I want to know how the current study is contextualised within current literature about collaboration between unpaid/family/friend caregiver and paid/unpaid others in relation to older adults in/leaving hospital. Some explanation/definitions of informal/unpaid caregivers would be helpful. Including research about single/primary and group/family caregivers would add depth to the paper. It may also be useful to look at existing literature around caregiver self-care during transitions of the cared-for person e.g., home to hospital; home to residential care; discharge to home.

RESPONSE 4: We have revised our introduction to provide a more thorough review of the literature and provide more background and definition related to the unpaid/family caregiver.

“However, unpaid/family caregivers include a wide range of individuals (family members, friends, neighbors, etc.), roles, and functions (self-care, self-management, social and emotional support) (5–8).”

“While there are a range of qualitative studies on unpaid/family caregiver experience of older adult hospital-to-home transitions (3), there are some critical gaps. Prior qualitative research has focused primarily on caregiver involvement in healthcare interactions (15–21), inefficient discharge processes (15,20,22–24), and tasks (e.g., medical and nursing) related to the older adult care (17,18,21,22,24–27). However, emerging evidence suggests a more complex collaborative process during hospital-to-home transitions (15,28).”

5. It would be educative for the authors to provide an example from their data analysis to demonstrate the analysis process.

RESPONSE 5: We have added Table 1 as an example of coding from data analysis.

6. A table listing the main themes and subthemes, with examples of quotation would provide a summary of the findings.

RESPONSE 6: We have added Table 3 to provide a summary of main categories with quotes.

7. The data is rich. The themes and subthemes are well-structured. It would be best to end each theme/subtheme with an interpretative comment, rather than with a participant quotation. This may mean rearranging rather than rewriting relevant theme/subthemes.

RESPONSE 7: Thank you for your comment. We have rearranged and added statements throughout consistent with this recommendation.

8. The discussion needs development. Please restate the purpose and summarise the key findings and then discuss each key finding in relation to the objective and relate these findings to prior studies. The authors need to show how the findings are consistent or contradictory with prior research – and as well as their own work. Where the results are contradictory, explain the possible reason.

RESPONSE 8: Thank you for the opportunity to improve our discussion. We have added key statements, including:

“Our study examined unpaid/family caregivers’ process of collaborating with others, including other family members, healthcare professionals, and community, social, and professional networks, during older adult hospital-to-home transitions. Our findings reveal a multi-stage transition process as caregivers support older adults through hospital-to-home transitions. Participants described various approaches to providing care and support, while supporting themselves, during the transition, which appear to involve more collaboration and utilization of networks than described in prior research.”

“While prior qualitative research on older adult hospital-to-home transitions has focused on the role of a single caregiver undergoing the transition process (15–27), our findings suggest a more complex team or “care network” structure of caregiving. This is consistent with caregiving research in other populations and efforts to reconceptualize caregiving to appreciate various roles, individuals, and dynamics in caregiving and caregiving networks (5–8).”

“Prior research has indicated a need for improved communication between healthcare professionals and caregivers and support during the transition (15,17,19,26). Our findings elucidate this issue, indicating that healthcare professional interactions with caregivers may primarily focus on how to address older adult needs during transitions. Conversations do not sufficiently address the support needs or challenges of caregivers.”

“Prior studies have indicated challenges in caregivers receiving assistance from family and friends, formal support services being implemented too late or ineffective, and lack of familiarity with available community services (17,18,21,22,24). Our findings indicate that social networks and professional services can serve an integral role in helping caregivers take on the many caregiver responsibilities while maintaining other roles, e.g., professional, social, household.”

“There is an opportunity to further leverage these and other community resources in future research and intervention, such as more purposeful matching of those in need with appropriate aging and caregiving resources (27,41–44).

“We must also assess the person’s financial situation and other social determinants of health that impact their transition (46).”

9. The limits of self-citation are evident in this section. The opening paragraph begins with, “While prior qualitative research with this population has focused on the role of a single caregiver undergoing the transition process (3), our findings suggest a more complex team or “care network” structure of caregiving.” (p. 20). As noted earlier, reference (3) is a self-citation not acknowledged as the authors’ prior work. The results should be situated within the broader literature to support the authors’ claims. For example, such literature should include previous studies that critique and unpack the “primary caregiver” model.

RESPONSE 9: Thank you. This comment has been addressed with response 8, especially in this statement: “While prior qualitative research on older adult hospital-to-home transitions has focused on the role of a single caregiver undergoing the transition process (15–27), our findings suggest a more complex team or “care network” structure of caregiving. This is consistent with caregiving research in other populations and efforts to reconceptualize caregiving to appreciate various roles, individuals, and dynamics in caregiving and caregiving networks (5–8).”

Reviewer 2

10. Methods- Line 101 page 4: This line is confusing. The authors should clarify if data saturation was reached for the parent larger study, or if the present analysis was a subset of individuals from the larger study and if data saturation was not used to decide when to end enrollment for the present analysis

RESPONSE 10: Thank you. We have clarified: “The present analysis occurred simultaneously with recruitment for the larger study, and we recruited until we reached data saturation for both the larger study (older adult interview data) and the present analysis (caregiver interview data) (33,34).”

11. The results section is poorly organized. This is perhaps the hardest section to follow. I think some structure in the initial paragraph could be added that would aid the reader in understanding how the results are presented (i.e. statements such as, “what follows are the themes/categories that were identified across the time period of transitioning from hospital to home”). 

RESPONSE 11: We have added table 3 as a summary of the results to help orient the reader. We have also adjusted transition statements in line with reviewer 1 comments. We have added some sentences to help the reader follow the results section:

“What follows is a description of categories across caregivers’ process of collaborating with others during older adult hospital-to-home transitions (shown in Figure 1). Table 3 provides a summary of main categories and example quotes.”

“In the later part of their transition experience, participants described their role in stepping back to encourage their independence in activities.”

“Conditions provide the context that impacted participants’ approach to caregiving and collaborating during the transition. Conditions that impacted participants’ approach to caregiving included finances, caregiver health, and older adult chronic conditions or repeated hospitalizations (Table 4).”

“Participants described consequences of their transition process differently based on their approach to caregiving and whether they were able to support progress versus planning to provide long-term care and support (see Figure 1).”

12. The three stages (identifying + learning, collaborating, supporting progress/planning) are well described and provide an interesting overview of the touchpoints with healthcare and community throughout this process. Seeing as there are subheadings under each stage, a summary paragraph would be helpful before proceeding into the next stage.

RESPONSE 12: We have provided table 3 as a summary. We also added some key statements throughout:

“What follows is a description of categories across caregivers’ process of collaborating with others during older adult hospital-to-home transitions (shown in Figure 1). Table 3 provides a summary of main categories and example quotes.”

“The way in which participants identified and learned their role seemed to inform their approach to collaborating with others to provide care and support, while also supporting self, in stage 2.”

“In the later part of their transition experience, participants described their role in stepping back to encourage their independence in activities.”

13. Discussion – line 456 + 464-466. This work also identifies an important implication for healthcare providers, namely in understanding the benefits that social/professional networks can provide caregivers, asking about these networks and referring to community resources when there is an identified lack. There is some discussion of promoting the role of care networks, but there is room to further expand on the role that healthcare plays in helping caregivers access these networks.

RESPONSE 13: Thank you for the great suggestion. We have added: “These findings also have implications for healthcare professionals, such as the need to understand the benefits that social/professional networks can provide caregivers, identifying opportunities to ask about these networks, and referring to community resources when there is an identified need.”

14. One the major limitations of this work is the lack of cultural diversity. This is referenced in the limitations section; however, the authors should consider exploring this limitation further. Care networks and their perceived impact/utility may different substantially between individualist and collectivist cultures.

RESPONSE 14: Thank you for the suggestion. We have revised to state: “Our sample is relatively small and lacks ethnic, cultural, and racial diversity. It will be critical to further explore how care networks and their perceived impact/utility may differ substantially such as between individualist and collectivist cultures.”

15. The authors could comment on the later timeline (30-60 days) of post-discharge interviews and how this might have impacted responses and themes as well as whether this was a strength or limitation of their study

RESPONSE 15: We have added: “Interviews took place approximately 30-60 days following the older adult’s hospital discharge, which allowed for a strong respective report of the entire transition experience. Variation in exact timing of interviews is both a strength and limitation, as it increases variation in responses that can contribute to both richness of data and variation in recount of experiences between participants. We did not interview any participants in the first 2 weeks post discharge (due to potential burden and required time to follow up and schedule the interview), which could be considered a limitation.”

16. The authors should comment on how the study taking place during the COVID-19 pandemic may have affected study findings. For example, in many parts of the world, there were restrictions imposed to caregivers coming to hospital due to public health restrictions which may have affected the experiences of caregivers

RESPONSE 16: Thank you. We have added: “Conduct of this study during the COVID-19 pandemic likely impacted study findings, such as access to resources and services, interactions with others, and visitation during hospitalization, which should be considered.”

17. It would be helpful for the reader if the authors were to delineate between ‘care in network’ and ‘care in social/professional networks’ earlier on. This comes up in the results section for ‘caring in social/professional networks’, but would be helpful to understand when describing ‘care in network’. Perhaps renaming to ‘shared caring in network’?

RESPONSE 17: Thank you for the suggestion. We have moved definitions up into Table 3 to provide distinction earlier on

---

## [Decision Letter · Decision Letter 1]

27 Aug 2024

Unpaid Caregivers’ Process of Collaborating with Others during Older Adult Hospital-to-Home Transitions: A Grounded Theory Study

PONE-D-23-42012R1

Dear Dr. Liebzeit,

We’re pleased to inform you that your manuscript has been judged scientifically suitable for publication and will be formally accepted for publication once it meets all outstanding technical requirements.

Kind regards,

Moustaq Karim Khan Rony, RN, MSS, MPH

Academic Editor

PLOS ONE

Additional Editor Comments (optional):

Reviewers' comments:

Reviewer's Responses to Questions

**Comments to the Author**

1. If the authors have adequately addressed your comments raised in a previous round of review and you feel that this manuscript is now acceptable for publication, you may indicate that here to bypass the “Comments to the Author” section, enter your conflict of interest statement in the “Confidential to Editor” section, and submit your "Accept" recommendation.

Reviewer #2: All comments have been addressed

2. Is the manuscript technically sound, and do the data support the conclusions?

Reviewer #2: Yes

3. Has the statistical analysis been performed appropriately and rigorously? 

Reviewer #2: Yes

4. Have the authors made all data underlying the findings in their manuscript fully available?

Reviewer #2: Yes

5. Is the manuscript presented in an intelligible fashion and written in standard English?

Reviewer #2: Yes

6. Review Comments to the Author

Reviewer #2: Summary

The authors have addressed the reviewer comments. The structure of the results section is more clear, and providing definitions of the different types of caring has strengthened the conceptual model.

Minor revision

Results – Line 437 onwards. The presentation of the conditions and consequences of caring is not as a strong as the presentation of the stages of caring through the transition. Specifically, the conceptual model (Figure 1) does not incorporate how conditions intersect with the stages – this very well may be across all stages, but may be helpful to note. Consequences are only presented for Stage 2 – caring on own, and Stage 3 – Planning to provide long-term care and support. It may be helpful to descriptively state in the results which stages the consequences and conditions apply to, to reinforce the visual graphic.

7. PLOS authors have the option to publish the peer review history of their article (what does this mean?). If published, this will include your full peer review and any attached files.

Reviewer #2: No

---

## [Editor Report · Acceptance letter]

9 Sep 2024

PONE-D-23-42012R1 

PLOS ONE

Dear Dr. Liebzeit, 

I'm pleased to inform you that your manuscript has been deemed suitable for publication in PLOS ONE. Congratulations! Your manuscript is now being handed over to our production team.

Kind regards, 

on behalf of

Mr. Moustaq Karim Khan Rony 

Academic Editor

PLOS ONE